# Inflammatory Changes and Composition of Collagen during Cervical Ripening in Cows

**DOI:** 10.3390/ani12192646

**Published:** 2022-10-01

**Authors:** Eigo Yamanokuchi, Go Kitahara, Kazuyuki Kanemaru, Koichiro Hemmi, Ikuo Kobayashi, Ryoji Yamaguchi, Takeshi Osawa

**Affiliations:** 1Laboratory of Theriogenology, Department of Veterinary Sciences, University of Miyazaki, Miyazaki 889-2192, Japan; 2Sumiyoshi Livestock Science Station, Field Science Center, Faculty of Agriculture, University of Miyazaki, Miyazaki 880-0121, Japan; 3Laboratory of Veterinary Pathology, Department of Veterinary Sciences, University of Miyazaki, Miyazaki 889-2192, Japan

**Keywords:** cervical ripening, cows, interleukin-8, mucus, picrosirius red staining, polymorphonuclear neutrophils, type I collagen

## Abstract

**Simple Summary:**

One cause of parturition abnormalities in cows may be the insufficient opening of the birth canal; however, the process by which the canal becomes flexible in cows is unknown. Herein, we observed changes in the cervix from late pregnancy to parturition to clarify the physiological changes related to cervical ripening during normal calving. Mucus and tissue samples from the cervix were collected from 41 cows at 30-day intervals from 200 to 260 days of gestation and at 7-day intervals from 260 days of gestation to parturition. The percentage of neutrophils in the mucus (PMN%) was calculated, and the concentration of interleukin (IL)-8, an inflammatory cytokine, was measured. The cervical tissues were observed under a microscope, and the percentage of type I collagen was calculated. The PMN% in the cervical mucus increased 5 weeks before delivery, peaking 1 week before calving; IL-8 was significantly increased at 295 days compared to that at 200 days of gestation. The percentage of type I collagen in the cervical tissue decreased significantly from 200 days to 274 days of pregnancy and continued to decrease thereafter until the week of calving. In conclusion, the cervix of cows begins to ripen at approximately 260 days of pregnancy.

**Abstract:**

Dystocia and stillbirths in cows pose a high risk of loss of both dams and fetuses, thereby resulting in high economic losses. One of the causes of these problems is birth canal abnormalities. Thus, to prevent these occurrences, it is necessary to understand the mechanisms underlying cervical ripening. Although physiological inflammatory responses and changes in collagen composition have been reported in humans and mice, related information is scarce for cows. We observed inflammatory changes and changes in the collagen composition in the cervix from late pregnancy to parturition to clarify some of the physiological changes associated with cervical ripening during normal calving in cows. Cervical mucus and tissue samples were collected from 41 Japanese Black cows at 200, 230, and 260 days of gestation and at 7-day intervals thereafter until parturition. The percentage of polymorphonuclear neutrophils (PMN%) in the mucus was calculated, and interleukin (IL)-8 concentration was determined by enzyme-linked immunosorbent assay. Blood samples were collected from the jugular vein, and leukocyte counts were determined. Picrosirius red-stained cervical tissue specimens were observed under a polarizing microscope, and the percentage of type I and type III collagen areas in the cervical tissue were calculated. The PMN% in cervical mucus was lowest at 200 days gestation (12–13 weeks before delivery), significantly increased 5 weeks before (21.7 ± 0.04), and was highest 1 week before calving (50.9 ± 0.04). IL-8 levels were increased at 295 days compared with those at 200 days of pregnancy (*p* < 0.05). No significant changes were observed in the white blood cell counts. The percentage of type I collagen in the cervical tissue reached a maximum (91.4 ± 0.02%) on day 200, significantly decreased after 274 days (3 weeks before calving), and continued to decrease thereafter until the week of parturition. There was no significant change in type III collagen levels. The results suggest that cervical ripening progresses when PMNs begin to infiltrate the cervix at around 260 days of gestation (5–4 weeks before parturition), IL-8, which increases at the end of pregnancy, mobilizes PMNs, and enhances inflammation, and that type I collagen changes are useful as an indicator of cervical ripening.

## 1. Introduction

Abnormal calving is a major factor driving reduced productivity. Prolonged gestation, excessive fetal size, and abnormal birth canal are all closely related to dystocia and stillbirth in cattle [1,2]. One of the pathophysiologies of dystocia due to birth canal abnormalities is the inadequate opening of the cervix. The cervix is a collagen-rich tissue located at the border between the uterus and vagina [3]. Normally, the cervix is tightly closed to prevent ascending infection and to retain the fetus in the uterus during pregnancy. However, as parturition approaches, the cervical canal softens and opens, allowing passage of the fetus during delivery. The sequence of processes by which the cervix softens for delivery is called cervical ripening [4]. Insufficient opening of the cervix may be caused by an abnormality in the cervical ripening mechanism [5]. When cervical ripening is incomplete, the cervix cannot dilate sufficiently, making delivery of the fetus difficult and resulting in dystocia [6]. In addition, dystocia and induced parturition in cattle may lead to placental retention and endometritis, which may adversely affect reproductive performance [7]. Therefore, evaluation of cervical ripening is important not only in predicting the timing of calving but also in preventing dystocia and improving postpartum productivity.

Degradation of collagen fibers and reduction of glycosaminoglycans, such as hyaluronic acid, have been observed during cervical ripening in humans and mice [8,9], and marked infiltration of inflammatory cells, such as neutrophils and macrophages, has been observed in the stroma [10,11,12,13]. Various studies have also shown that activation of inflammatory cells in humans and guinea pigs results in the secretion of inflammatory cytokines, such as interleukin (IL)-8, which induces neutrophil chemotaxis and degranulation [13,14,15]. Mouse studies have shown that, as a result, various proteases, such as the collagen-degrading matrix metalloproteinases (MMPs) and hyaluronan synthases (HASs), are released and alter the extracellular matrix. Cervical ripening is thought to be caused by the remodeling of the extracellular matrix due to physiological inflammation [13,15]. Type I collagen is the major component responsible for the mechanical strength of the cervical canal, and in humans and mice, a decrease in the ratio of type I collagen and an increase in the ratio of type III collagen in the cervix at term may induce cervical softening before parturition [16,17]. Compared to humans and mice, less is known about the mechanism of cervical ripening in cattle. Although several reports have indicated that the mRNA levels of inflammatory cytokines in cervical tissue increase from 185 to 275 days of gestation (5–6 days before parturition) [18] and that neutrophils in cervical mucus increase from 8 days before parturition [19] in cattle, there have been no reports of continuous observation of cytokines in cervical mucus at the protein level or of changes in collagen composition in cervical tissue prior to delivery.

In human obstetrics, the Bishop score [20], which is based on the position of the uterine opening, cervical patency, firmness, length, and position of the fetal head, is used to assess cervical ripening to predict preterm and difficult labor and to improve the success rate of induced labor. However, no system in bovine practice can objectively and comprehensively evaluate cervical ripening, such as the Bishop score. If signs of abnormal calving in cattle can be detected in advance, it may be possible to save newborns and prevent postpartum diseases by accelerating cervical ripening. Therefore, there is a need to establish, firstly, a procedure to accelerate cervical ripening and, secondly, an indicator to evaluate this procedure. To understand a part of the mechanism of cervical ripening in cattle, it is necessary to elucidate changes in normal cervical tissue during the process of cervical ripening.

In this study, we observed changes in inflammatory cytokine and neutrophil dynamics in the cervical mucus and collagen in cervical tissue to clarify the physiological inflammatory profile associated with cervical ripening during normal calving in cattle.

## 2. Materials and Methods

### 2.1. Animals

Forty-one Japanese Black cows (age: 10.4 ± 0.6 years old, parity: 8.5 ± 0.6; mean ± SEM) that calved normally at Sumiyoshi Field Center, Faculty of Agriculture, University of Miyazaki, between September 2019 and September 2021 were used as the study objects. All animals were artificially inseminated with Japanese Black semen and were housed in free-range barns, fed hay (Italian ryegrass, guinea grass, and rose grass), Italian ryegrass wrap silage, and corn silage ad libitum. Cows were kept in the calving room from 275 days of pregnancy until four weeks postpartum, where they received 4 kg/day of compound feed (corn, barley, sorghum, wheat bran, rice bran, soybean oil cake, rapeseed oil cake; Cowgreat^TM^ AP, Nosan Corporation, Yokohama, Japan) in addition to the above roughage. The mean number of gestation days was 292.8 ± 0.9 days, and the mean birth weight of the calves was 33.4 ± 0.8 kg. None of the animals had clinical signs of peripartum diseases and showed abnormal values in complete blood count parameters during the study period.

This study was approved by the Institutional Animal Care and Use Committee, University of Miyazaki (2016-009, 2021-041).

### 2.2. Study Design

Blood and cervical mucus samples were collected from cows on days 200, 230, 260, and every 7 days after day 260 (all ±3 days), and cervical tissue samples were collected on gestation days 200, 260, 274, and 288 (all ±3 days) of pregnancy, and repeated every 7 days thereafter until calving (Figure 1).

### 2.3. Cervical Mucus Collection

A hand wearing a glove (Fujihira Kogyo Co., Ltd., Tokyo, Japan) disinfected with 70% ethanol was inserted into the vulva and disinfected with 0.02% mono, bis (trimethylammonium methylene chloride)-alkyl toluene, and a small amount (10 g) of cervical mucus was collected with the fingers to fit inside the palm. A portion of the collected mucus was promptly applied to a glass slide and fixed with Cytokeep II (Alfresa Pharma Corporation, Osaka, Japan). The smear was stained with Diff-Quik (Alfresa Pharma Corporation, Osaka, Japan). The remaining mucus was placed in 1.5 mL microtubes, transported at 4 °C within 5 h, and stored at −80 °C until processing, as described below.

#### 2.3.1. Percentage of Polymorphonuclear Neutrophils (PMN%) of Cervical Mucus

For the 41 cows, 15 cervical mucus samples were collected between September 2020 and September 2021. Four hundred nucleated cells were observed in the 200–400× field of view of an upright microscope, and PMN% was calculated as the percentage of PMNs in the smear samples.

#### 2.3.2. Protein Analysis of Cervical Mucus

##### Cervical Mucus Processing and Protein Quantification

The mucus from 36 of the 41 animals was tested because some of the samples of the five cows were missing. Cervical mucus stored at −80 °C was thawed, and 0.1 g was mixed well with 0.9 g of phosphate-buffered saline (PBS). The mixture was incubated at room temperature for 1 h, and the supernatant was collected after centrifugation at 4 °C for 30 min at 20,630× *g*. The protein concentration in the supernatant was determined by measuring the absorbance at 280 nm with a spectrophotometer (Nanodrop-8000, Thermo Fisher Scientific, Waltham, MA, USA).

##### Determination of IL-8 Concentrations in Cervical Mucus

As in previous reports [21], the bovine IL-8 (CXCL8) ELISA BASIC kit 3114-1H-6 (MABTECH, Nacka Strand, Sweden) was used to determine IL-8 concentrations according to the manufacturer’s instructions. Briefly, antibodies were bound to 96-well plates overnight; 200 µL/well of PBS containing 0.05% Tween 20, and 0.1% bovine serum albumin was added, incubated at 25 °C for 1 h, and the plates were then blocked and washed five times with PBS containing 0.05% Tween 20. Subsequently, 100 µL of the sample and standard solution per well was added, incubated at 25 °C for 2 h, and washed as described above. Then, 100 µL of detection antibody per well was added and incubated at 25 °C for 1 h, followed by washing again, as described above. Next, 100 µL streptavidin–horseradish peroxidase was added to each well and incubated at 25 °C for 1 h. After the fourth wash, 100 µL of TMB substrate (KPL SureBlue™ TMB Microwell Peroxidase Substrate; Sera Care Life Science, Milford, MA, USA) was added at 100 µL per well and incubated for 15 min. Finally, 100 µL of 0.2 M H₂SO₃ was added to each well to stop the reaction. The absorbance at 450 nm was measured within 15 min, and the absorbance at 630 nm was subtracted. Absorbance was measured using a microplate reader (Multiskan GO, Thermo Fisher Scientific, Waltham, MA, USA), and the results were analyzed using the SkanIt™ software for microplate readers (SkanIt Software 3.2.0.35 RE for Multiskan GO, Thermo Fisher Scientific, Waltham, MA, USA).

#### 2.3.3. Collagen Analysis of Cervical Tissue

##### Sample Collections and Processing

Cervical tissues from 28 of the 41 test cows were sampled because the remaining 13 animals were not present at all the sampling points and/or biopsy samples could not be taken in sufficient quantities for the analysis due to the narrow cervix. Following the protocol of a previous report [22], a portion of cervical tissue was collected using a biopsy punch (Biopsy Trepan, Kai Corporation, Tokyo, Japan). The cervical tissues were fixed in 4% paraformaldehyde phosphate buffer and embedded in paraffin. After paraffin embedding, sections were prepared at 5 µm and stained with the Picro-Sirius Red Stain Kit (Scy Tec Laboratories, Logan, UT, USA) according to the manufacturer’s instructions [23]. Briefly, tissue was deparaffinized with xylene and ethanol, rehydrated with distilled water for 15 min, covered with Picro-Sirius Red solution, and incubated for 60 min to prevent drying. The tissue was then rinsed twice with 0.5% acetic acid, dehydrated with ethanol, permeabilized with xylene, and sealed.

##### Image Analyses

When the picrosirius red-stained specimens were observed under a polarizing microscope, type I collagen was distinguished as red to yellow and type III collagen as green areas [23]. The specimens were photographed under a polarizing microscope with a 200× field of view to capture the entire area from the epithelium to the stroma. The hue of type I collagen was defined as 2–51 and 230–256, and that of type III collagen was defined as 52–128, according to the method of Rich et al. [24]. The captured images were analyzed and the percentage of type I and type III collagen in the cervical tissue was calculated using ImageJ (Version 1.53k, National Institute of Health, Bethesda, MD, USA).

### 2.4. Leukocyte Counts in Peripheral Blood

Leukocyte counts in peripheral blood was analyzed for 15 of the 41 cows. Blood samples were collected from the jugular vein using a vacuum blood collection tube containing EDTA-2Na (Terumo Corporation, Tokyo, Japan) and a 21 g blood collection needle. Blood was transported at 4 °C within 5 h of collection, and the white blood cell count was determined using an automated hematology analyzer (Celltac α, MEK-6550, Nihon Kohden, Tokyo, Japan).

### 2.5. Statistical Analysis

All results were compared in terms of the number of days of pregnancy, with the day of insemination being 0 for all items. For PMN%, IL-8, and collagen, the week of calving was defined as w0 (0–6 days before calving), 1 week before (w-1), 2 weeks before (w-2), 3 weeks before (w-3), 4 weeks before (w-4), 5 weeks before (w-5), 9–8 weeks before (w-9 to -8), and 13–12 weeks before calving (w-13 to -12).

RStudio (Version 1.3.1056, RStudio, Boston, MA, USA) was used for statistical analysis. A two-way analysis of variance with repetition was performed on all items, and Tukey-Kramer tests were performed as post-tests for those items that showed significant differences (*p* < 0.05). All results are expressed as mean ± standard error of the mean.

## 3. Results

### 3.1. Cervical Mucus

#### 3.1.1. PMN%

PMN% increased (*p* < 0.05) after 260 days (18.5 ± 0.02) compared to that in the first sampling (200 days of pregnancy), reaching a peak (50.2 ± 0.02) at 281 days (Figure 2). There was also a significant increase (*p* < 0.05) in w-5 (21.7 ± 0.04) and a peak (50.9 ± 0.04) in w-1 compared to that in w-13 to -12; the first sample was collected (Figure 3).

#### 3.1.2. Determination of Total Protein and Cytokine Concentrations

Protein concentrations in the cervical mucus showed an increasing trend (*p* = 0.08).

The IL-8 concentration in the cervical mucus was significantly different at 295 days (990.5 ± 236.9 pg/mL) versus 200 (326.3 ± 85.2 pg/mL), 230 (335.8 ± 98.5 pg/mL) and 260 (296.1 ± 84.6 pg/mL) days of pregnancy (Figure 2). IL-8 increased from w-13 to -12 (333.3 ± 94.0 pg/mL) to w-2 (650.9 ± 145.8 pg/mL) and w-1 (715.7 ± 136.9 pg/mL) in the initial collection, but with no significant difference (*p* = 0.11; Figure 3).

### 3.2. Changes in Collagen Composition in Cervical Canal Tissue

The regions of type I and type III collagen from 200 to 288 days of pregnancy in a representative cow are shown in Figure 4. Type I collagen decreased after 260 days (85.6 ± 0.02%) compared to that at 200 days (91.4 ± 0.02%) of pregnancy (*p* < 0.05; Figure 5). A decrease was also observed in w-3 (76.1 ± 0.05%) and onwards compared to that in w-13 to -12 (*p* < 0.05; Figure 6). Type III collagen showed no significant difference throughout the study period (*p* = 0.36; Figure 6).

### 3.3. Leukocyte Count in Peripheral Blood

Leukocyte counts (mean, median, range) in peripheral blood were day 200 (66.8, 66, 51–90), day 230 (63.5, 64, 45–83), day 260 (62.0, 62, 47–76), day 267 (62.7, 63, 40–88), day 274 (62, 41–83), day 281 (61.3, 61, 43–79), day 288 (64.5, 63, 46–79), and day 295 (70.7, 70, 56–88). No significant changes in leukocyte count were observed in the peripheral blood.

## 4. Discussion

In the present study, we observed and analyzed the inflammatory changes in cervical mucus and cervical tissue of pregnant cows to gain a better understanding of the mechanism of cervical ripening in cows during late pregnancy. Our results suggest that cervical ripening progresses with PMN infiltration from around 260 days of gestation (5 to 4 weeks before calving). In addition, IL-8, which increased two weeks before calving, mobilized PMNs, which enhanced the inflammation and decreased type I collagen in the cervical canal tissue, suggesting the progression of cervical ripening.

An increase in PMNs and IL-8 in the cervical mucus was observed as calving approached, suggesting that PMNs infiltrate the cervical stroma from the blood vessels and migrate with mucus into the cervical lumen [13]. In the present study, IL-8 levels increased two weeks before parturition, and PMN% peaked one week before parturition. Thus, as in previous reports in humans, IL-8 levels were increased by suppressing the cervical progesterone receptor, suggesting that progesterone and its receptor may be involved in cervical ripening [25,26]. In the future, the relationship between IL-8 variation and progesterone should be clarified by immunostaining of the receptor and measurement of the blood progesterone levels to elucidate the mechanism of cervical ripening further.

The type I collagen in the cervical tissue decreased as calving approached. Type I collagen makes up the majority of the extracellular matrix of cervical tissue and provides strength to the cervix. Cervical ripening in humans is thought to reflect cervical softening due to a decrease in the percentage of type I collagen in the cervical tissue [17]. However, some reports have indicated that cervical softening in humans and mice is characterized by increased collagen solubility, with no change in total collagen content [27], suggesting that the decrease in collagen in cervical tissue may not be quantitative but qualitative, such as a decrease in collagen cross-link density [28]. Although there were no significant changes in type III collagen in the present study, it has been reported that type III collagen is present in the early stages of wound healing [29]. Therefore, both the quantitative and qualitative changes, along with their roles, need to be examined. In addition, recent reports have indicated that smooth muscle cells play an active role in cervical ripening, as they account for more than 50% of the smooth muscle cells in the uterus and uterine side of the cervix in humans [30], tissue changes during inflammation are not due to increased or decreased collagen content or cross-linking, but rather to the fact that IL-1β decreases cervical fibroblasts [31]. The function of smooth muscle cells and fibroblasts in cervical ripening, as well as collagen changes in cervical tissue, should be investigated in the future.

Although an increase in the number of leukocytes in peripheral blood on the day of parturition has been reported [32], no significant increase was observed in the present study. Moreover, in the present study, we did not perform frequent sampling a few hours before parturition; therefore, changes over time during this period are unknown. However, as inferred from previous reports in cattle [33], the inflammatory changes in the peripheral blood are likely acute. As blood cortisol levels increase during parturition [34], the inflammatory changes in peripheral blood are due to stress during parturition. Thus, cervical ripening is considered a local response unrelated to systemic inflammatory changes, which is supported by the findings of Monsanto et al. [35].

In cattle, fetal displacement, such as flexion [36], narrowing of the dam’s pelvic cavity [37], and fetal overgrowth [2], have been suggested to be associated with dystocia; however, little is known about cervical ripening in comparison to these factors. In addition, prolonged gestation in cattle is a risk factor for dystocia [1], and since the average gestation period of Japanese Black cows has increased in recent years [1,38,39], it is necessary to take further measures to prevent dystocia. A better understanding of the mechanism of cervical ripening may help to prevent dystocia. In humans, cervical ripening is promoted by the administration of cervical dilators and mechanical dilation [40], and applying similar methods to cows with inadequate cervical ripening may reduce the risk of dystocia. The peptide hormone relaxin plays a key role in the widening of the pubic bone during labor and extracellular matrix remodeling [41,42]. Continuous intravenous infusion of relaxin resulted in a decrease in progesterone secretion in late pregnant beef heifers [43]. However, the role of relaxin in cervical ripening in cattle is unknown and needs further investigation.

Although prolonged gestation is a concern in cattle, research in humans has mostly focused on preventing premature births. In human obstetrics, the Bishop score is used to score the texture and patency of cervical tissue, which is used to predict preterm delivery [20]. However, cervical behavior is dictated by complex biomechanics and involves multiple pregnancy tissues, and to understand the complex biomechanics that informs its behavior, the intricacies of the related events at the intersection of the cervix, membranes, and uterus must be clarified [44]. In the future, accurate monitoring and evaluation of the progress of cervical ripening based on the biomechanics and clinical application of these findings will be an effective tool in bovine practice. In humans, shortening of the cervical canal length during pregnancy is known to be associated with preterm delivery [45,46,47]. Various ILs and MMPs have been implicated in cervical shortening in humans, IL-6 and IL-8 in the cervical mucus are predictors of impending preterm delivery [48], and increased granulocyte elastase in cervical secretions is an independent predictor of preterm delivery before 34 weeks or earlier [49]. The profiles of IL-8 in the cervical mucus of late pregnant cows in this study may allow this cytokine to be used as a predictor of the timing of the onset of parturition in cattle.

This is the first study to observe IL-8 protein levels in cervical mucus prior to parturition in cattle. Overall, we observed the changes in cervical mucus cytokines and cervical tissue collagen over a period of approximately three months before parturition. In the future, we should clarify the details of cervical ripening in cattle, including analyzing the detailed changes over a short period of time, such as the day before and the day of calving, and establish a system to evaluate cervical ripening by comparing it with premature delivery, prolonged gestation, and dystocia. It is expected that the evaluation of the progress of cervical ripening will be utilized in the bovine practice as a tool to estimate the timing of calving and to predict dystocia.

## 5. Conclusions

In the present study, we observed the inflammatory changes in the cervical mucus, tissues, and peripheral blood during cervical ripening in cattle. Overall, we found that cervical ripening was initiated when PMNs began to infiltrate the cervix at approximately 260 days of gestation (5–4 weeks before parturition), IL-8 increased toward the end of pregnancy, and the percentage of type I collagen in the cervical tissue significantly decreased after 274 days (3 weeks before calving), and continued to decrease thereafter until the week of parturition.

## Figures and Tables

**Figure 1 animals-12-02646-f001:**
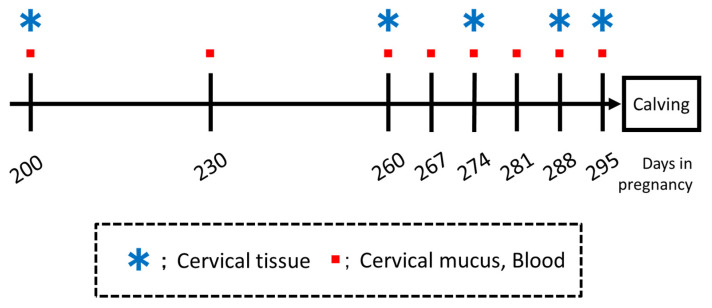
Experimental design.

**Figure 2 animals-12-02646-f002:**
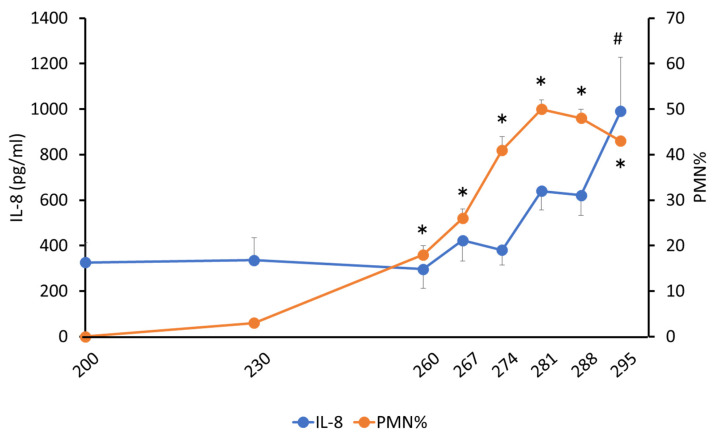
Interleukin-8 and polymorphonuclear neutrophil % in bovine cervical mucus with days of pregnancy. # *p* < 0.05 compared with 200 days in IL-8; * *p* < 0.05 compared with 200 days in PMN%.

**Figure 3 animals-12-02646-f003:**
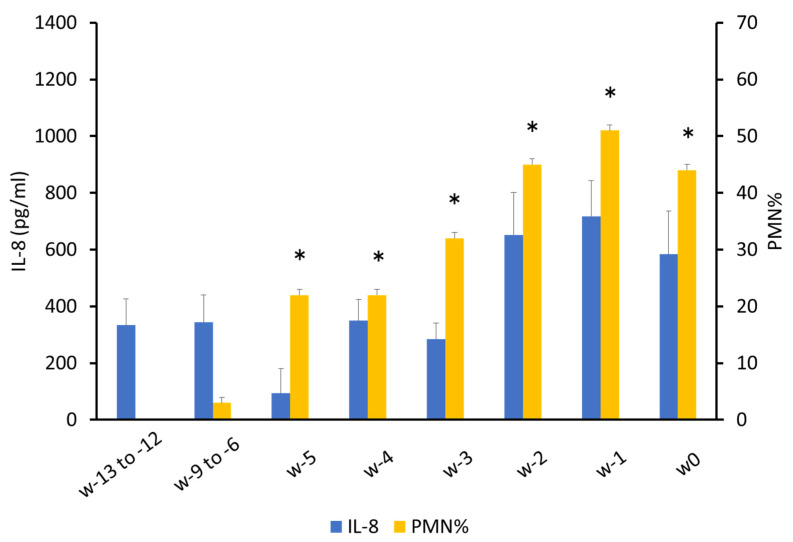
Interleukin-8 and polymorphonuclear neutrophil % in bovine cervical mucus with weeks to calving. * *p* < 0.05 compared with w-13 to -12 in PMN%.

**Figure 4 animals-12-02646-f004:**
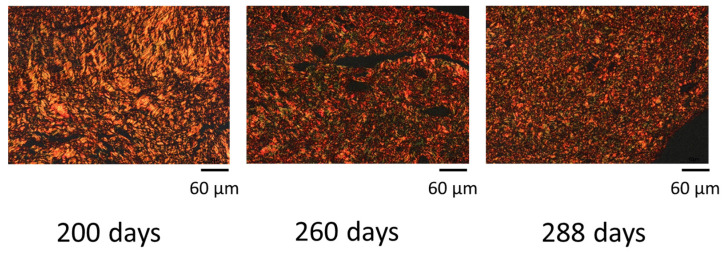
Images of picrosirius red-stained cervical tissues in a cow at 200, 260, and 288 days of pregnancy. Example of a representative cow. Red-yellow: Type I collagen; Green: Type III collagen.

**Figure 5 animals-12-02646-f005:**
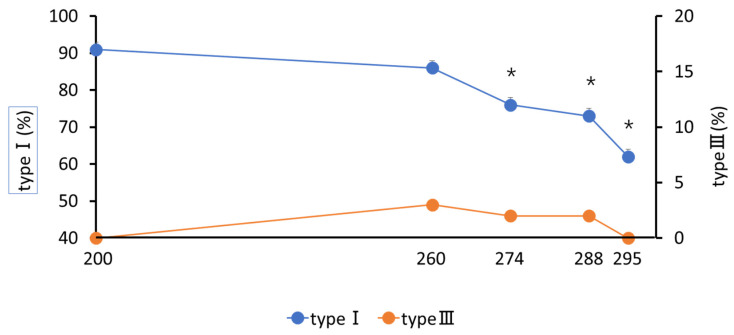
Changes in the percentage of cervical tissue type I and type III collagen with days of pregnancy. * *p* < 0.05 compared with 200 days in type I.

**Figure 6 animals-12-02646-f006:**
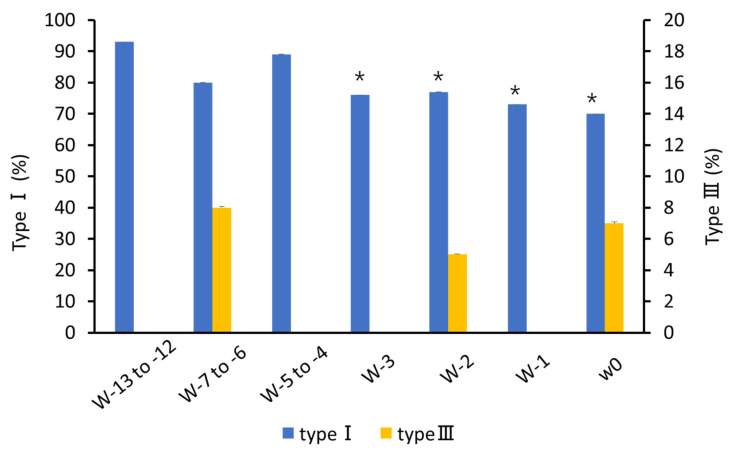
Changes in the percentage of cervical tissue type I and type III collagen with weeks to calving. * *p* < 0.05 compare with w-13 to -12 in type I.

## Data Availability

The data presented in this study are available upon request from the corresponding author. The data are not publicly available due to privacy.

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
