# Peer review of "Inflammatory Changes and Composition of Collagen during Cervical Ripening in Cows"

_animals, 2022, doi:10.3390/ani12192646_

Round 1

Reviewer 1 Report

It has been a great pleasure for me to read your innovative paper on cervical ripening. Very few papers have been published on this very practical aspect for the farmes and practitionners.  Thanks for the done work.

I have some minor remarks

Line 140 : every 7 days after day 260 could be better

Line 141 : You write the mean number of gestation days was 292.8 ± 0.9 days but how many cows have a 295 days of pregnancy lenght ?

Line 158 what kind of coloration have you used ?

Line 245 In figure 2 the concentration of IL-8 show an increase at day 295 but a decrease in figure 3. Of course, the period of evaluation is different.  What’s important to understand ?

Line 293 is it right to think that it’s the decrease of progesterone who is involved in the increase of IL-8 concentration ?

Line 327 could you suggest the role of relaxin in cervical ripening ?

Line 347 Have you read the paper of Feltovitch ?

Author Response

Line 140: every 7 days after day 260 could be better

Response → Thank you for your valuable comments. We have revised it according to the reviewer’s suggestion.

Line 141: You write the mean number of gestation days was 292.8 ± 0.9 days but how many cows have a 295 days of pregnancy length?

Response → Four cows calved at 295 days. Twenty-two cows calved at less than 295 days and 10 cows calved at 296 or more days.

Line 158: what kind of coloration have you used?

Response → The smear was stained with Diff-Quik as stated in Line 153.

Line 245: In figure 2 the concentrations of IL-8 show an increase at day 295 but a decrease in figure 3. Of course, the period of evaluation is different.  What’s important to understand?

Response → As the Reviewer points out, the period of evaluation is different. In order to understand when the signs of cervical ripening begin, we thought it would be necessary to observe how many days before the calving, as well as how many days of pregnancy. Therefore, we presented the results at two different time points. The results of the present study suggest that IL-8 concentrations in cervical mucus increase in the final stage of pregnancy, but may peak several days before calving. The details of the profiles of IL-8 concentrations in normal calving and the determination of normal values will be clarified in further studies in the future.

Line 293: is it right to think that it’s the decrease of progesterone who is involved in the increase of IL-8 concentration?

Response → As described in this paragraph, although there is evidence in previous studies that suppressing the physiological effects of progesterone in humans increased IL-8 levels, it is unclear whether the same is true in cattle, and this should be observed in the future.

Line 327: could you suggest the role of relaxin in cervical ripening?

Response → The following sentences has been added in Discussion. ‘The peptide hormone relaxin plays a key role in widening of the pubic bone during labor and extracellular matrix remodeling [Malone et al., 2017; Jelinic et al., 2018]. Continuous intravenous infusion of relaxin resulted in a decrease in progesterone secretion in late pregnant beef heifers [Smith et al., 1997]. However, the role of relaxin in cervical ripening in cattle is unknown and needs further investigation.’

Line 347: Have you read the paper of Feltovitch?

Response → The following sentences has been added in Discussion. ‘However, cervical behavior is dictated by complex biomechanics and involves multiple pregnancy tissues, and to understand the complex biomechanics that informs its behavior, the intricacies of the related events at the intersection of the cervix, membranes and uterus must be clarified [Feltovich, 2019].’

Reviewer 2 Report

An easy to follow manuscript with some slight recommendations as follows:

Line 68: define the birth canal; the cervix forms part of the birth canal

Line 111-114: Rephrase this aim

Line 132: …and showed no abnormal values…

Line 138-139: “Blood and cervical mucus samples were collected from cows on days 138 200, 230, 260, and 267 (all ± 3 days),”. This statement disagrees with what is shown in Figure 1

Line 145 and 146: Mention what antiseptic was used on the glove and on the vulva

Line 162: Explain why only mucus from 36 cows was evaluated

Line 166-169: What is meant by the comment, “….until the next item was measured”? Was the supernatant stored and reused?

Line 194: Explain why only cervical tissues from 28 of the 41 test cows were sampled

Line 218: Why were only 15 cows blood sampled. Is this a typing error. Should it not be 41 cows?

Line 272-274: Provide mean, median and range for leucocyte count on each blood collection day

Line 312-315: Please rephrase this statement

Reference [25] missing in text.

Author Response

Line 68: define the birth canal; the cervix forms part of the birth canal

Response → Thank you for pointing out our vague expression on the birth canal. The phrase in Line 70, ‘the cervical canal (i.e., the birth canal)’ has been revised to ‘the cervix’.

Line 111-114: Rephrase this aim

Response → The phrase, ‘To evaluate cervical ripening’ has been revised to ‘To understand a part of the mechanism of cervical ripening in cattle’.

Line 132: …and showed no abnormal values…

Response → The sentence here means, ‘None of the animals showed abnormal values…’.

Line 138-139: “Blood and cervical mucus samples were collected from cows on days 138 200, 230, 260, and 267 (all ± 3 days),”. This statement disagrees with what is shown in Figure 1

Response → Thank you for your comment. It is our mistake. The phrase has been revised to “Blood and cervical mucus samples were collected from cows on days 200, 230, 260, and every 7 days after day 260 (all ± 3 days),”.

Line 145 and 146: Mention what antiseptic was used on the glove and on the vulva

Response → Mentioned.

Line 162: Explain why only mucus from 36 cows was evaluated

Response → Explained.

Line 166-169: What is meant by the comment, “….until the next item was measured”? Was the supernatant stored and reused?

Response → This phrase was incorrect. All samples were subjected to measurement in a single freeze-thaw.

Line 194: Explain why only cervical tissues from 28 of the 41 test cows were sampled

Response → Explained. In part, the cows were not present due to grazing at the time of sample collection. In addition, biopsy samples could not be taken in sufficient quantities for the analysis due to the narrow cervix.

Line 218: Why were only 15 cows blood sampled. Is this a typing error. Should it not be 41 cows?

Response → Leukocyte counts in peripheral blood was analyzed for 15 cattle, part of the 41 cows. We decided this sample size because this would be a sufficient number to observe the leukocyte count profiles.

Line 272-274: Provide mean, median and range for leucocyte count on each blood collection day

Response → Provided.

Line 312-315: Please rephrase this statement

Response → Thank you for your comment on our mistake. The phrase, “… the smooth muscle cells in the endometrium and …” has been revised to “… the smooth muscle cells in the uterus and …”.

Reference [25] missing in text.

Response → Thank you for your comment. Cited reference, [25], has been added in the text.